# Isogenic Human-Induced Pluripotent Stem-Cell-Derived Cardiomyocytes Reveal Activation of Wnt Signaling Pathways Underlying Intrinsic Cardiac Abnormalities in Rett Syndrome

**DOI:** 10.3390/ijms232415609

**Published:** 2022-12-09

**Authors:** Kwong-Man Ng, Qianqian Ding, Yiu-Lam Tse, Oscar Hou-In Chou, Wing-Hon Lai, Ka-Wing Au, Yee-Man Lau, Yue Ji, Chung-Wah Siu, Clara Sze-Man Tang, Alan Colman, Suk-Ying Tsang, Hung-Fat Tse

**Affiliations:** 1Cardiology Division, Department of Medicine, Li Ka-Shing Faculty of Medicine, The University of Hong Kong, Hong Kong SAR, China; 2Center for Translational Stem Cell Biology, Hong Kong SAR, China; 3School of Life Sciences, The Chinese University of Hong Kong, Hong Kong SAR, China; 4Department of Surgery, Li Ka-Shing Faculty of Medicine, The University of Hong Kong, Hong Kong SAR, China; 5Dr. Li Dak-Sum Research Centre, The University of Hong Kong, Hong Kong SAR, China; 6Harvard Department of Stem Cells and Regenerative Biology, Cambridge, MA 02138, USA; 7State Key Laboratory of Agrobiotechnology, The Chinese University of Hong Kong, Hong Kong SAR, China; 8The Institute for Tissue Engineering and Regenerative Medicine (iTERM), The Chinese University of Hong Kong, Hong Kong SAR, China; 9Hong Kong-Guangdong Stem Cell and Regenerative Medicine Research Centre, The University of Hong Kong and Guangzhou Institutes of Biomedicine and Health, Hong Kong SAR, China; 10Heart and Vascular Center, The University of Hong Kong-Shenzhen Hospital, Shenzhen 518053, China

**Keywords:** Rett syndrome, *MeCP2 mutation*, intrinsic cardiac abnormality, Wnt/β-catenin signaling pathway

## Abstract

Rett syndrome (RTT) is a severe neurodevelopmental disorder caused by *MeCP2 mutations*. Nonetheless, the pathophysiological roles of *MeCP2 mutations* in the etiology of intrinsic cardiac abnormality and sudden death remain unclear. In this study, we performed a detailed functional studies (calcium and electrophysiological analysis) and RNA-sequencing-based transcriptome analysis of a pair of isogenic RTT female patient-specific induced pluripotent stem-cell-derived cardiomyocytes (iPSC-CMs) that expressed either *MeCP2^wildtype^* or *MeCP2^mutant^* allele and iPSC-CMs from a non-affected female control. The observations were further confirmed by additional experiments, including Wnt signaling inhibitor treatment, siRNA-based gene silencing, and ion channel blockade. Compared with *MeCP2^wildtype^* and control iPSC-CMs, *MeCP2^mutant^* iPSC-CMs exhibited prolonged action potential and increased frequency of spontaneous early after polarization. RNA sequencing analysis revealed up-regulation of various Wnt family genes in *MeCP2^mutant^* iPSC-CMs. Treatment of *MeCP2^mutant^* iPSC-CMs with a Wnt inhibitor XAV939 significantly decreased the β-catenin protein level and *CACN1AC* expression and ameliorated their abnormal electrophysiological properties. In summary, our data provide novel insight into the contribution of activation of the Wnt/β-catenin signaling cascade to the cardiac abnormalities associated with *MeCP2* mutations in RTT.

## 1. Introduction

Rett syndrome (RTT) is a severe X-linked dominant neurodevelopmental disorder [1] that mainly affects females [2] and has a prevalence of 57–100 per million population [2,3,4]. Sporadic cases have been reported in males [5]. Clinically, RTT is characterized by neurodevelopmental regression, mental retardation and learning disability, as well as the loss of speech and manual dexterity [2,3,4]. Typically, male patients with RTT exhibit a more severe phenotype with early mortality, whereas females may survive beyond middle age [2,3,4]. Nonetheless, the life expectancy of female patients is reduced and associated with an increased incidence of sudden death, accounting for approximately one-quarter of all their deaths [6]. Although sudden death in RTT is likely attributed to multiple causes, a significant proportion is believed to be cardiac in origin. Prior studies showed that a prolonged QT interval on electrocardiogram was evident in a significant proportion of affected patients [6,7], although the mechanisms of QT prolongation and sudden death in RTT remain unclear.

Mutation in the Methyl-CpG Binding Protein 2 (*MeCP2*) gene is present in more than 95% of RTT cases [8]. Functionally, MECP2 is a nucleoprotein that binds to methylated DNA and is believed to play an important role in the regulation of gene transcriptions [9]. *MeCP2* knock-out mice have recapitulated a considerable range of neurological phenotypes of RTT, including motor impairment and stereotypic forelimb motions [10,11,12,13]. Nonetheless, there are conflicting data on the cardiac phenotypes in *MeCP2* null mice. Using *MeCP2* null mice and nervous-system-specific conditional *MeCP2* knock-out (NKO) mice, McCauley et al. [14] demonstrated that deletion of *MeCP2* in the nervous system alone was sufficient to cause QT prolongation and an increased susceptibility to ventricular tachycardia. Interestingly, they also observed increased persistent sodium currents (I_Na_) in ventricular myocytes isolated from *MeCP2* null mice and *MeCP2* NKO mice compared with wildtype controls [14]. 

On the contrary, recent studies from Hara et al. failed to demonstrate any cardiac functional abnormalities or QT prolongation in *MeCP2* null mice, despite the fact that both experiments used the same strain of mice carrying the same mutant allele [15]. The exact reasons for these different observations are unclear but may be due to differences between the two studies in animal age. Moreover, there are substantial differences in the profile of cardiac ion channels between humans and rodents, so the *MeCP2* null mouse model may not fully recapitulate the pathophysiological mechanisms of QT prolongation related to MECP2 protein deficiency in humans with RTT.

To date, adult human somatic cells have been reprogrammed with well-defined factors into human induced pluripotent stem cells (iPSCs) [16,17] and can be further differentiated into various cell types, such as neurones and cardiomyocytes (CMs), to model genetic diseases. Patient-specific isogenic iPSC lines have been used to model the neuronal defects of RTT [17]. For X-linked mutations, as in RTT, the X-chromosome inactivation status is unchanged during reprogramming. Reprogramming of pre-sorted mosaic somatic cells from a female heterozygous carrier may allow us to generate a pair of isogenic iPSC lines that are genetically identical except in the X-chromosome inactivation status of the mutant gene [16,17].

In this study, we utilized a pair of isogenic wildtype (*MeCP2^wildtype^*) and mutant *(MeCP2^mutant^*) iPSCs from a female patient with RTT [18] and differentiated them into functional CMs to investigate the potential mechanisms of QT prolongation and sudden death in RTT. Since these lines are genetically identical except for their ability to produce the functional MECP2 protein, we were able to study the direct effects of *MeCP2* mutation on the development of cardiac dysfunction associated with RTT. Using these isogenic *MeCP2^wildtype^* and *MeCP2^mutant^* iPSC-CMs as a platform, we demonstrated that altered Wnt signaling due to MECP2 protein deficiency is associated with altered cardiac ion channel expression and cellular electrophysiology that underlie the QT prolongation and sudden death in RTT.

## 2. Results

### 2.1. MeCP2 Mutation Caused Increased Irregular Contraction of iPSC-CMs

As shown in Figure 1A, there was no change to X-chromosome inactivation status during cardiac differentiation, as both *MeCP2^wildtype^* iPSCs and iPSC-CMs as well as *MeCP2^mutant^* iPSCs and iPSC-CMs expressed the corresponding wildtype (158 bp) or the mutant (126 bp) *MeCP2* allele, respectively. Immunostaining analysis further demonstrated a reduction in mature MECP2 proteins in the nucleus of *MeCP2^mutant^* iPSCs and iPSC-CMs (Figure 1B). Moreover, the *MeCP2* mutation did not affect cardiac differentiation. During cardiac differentiation, both wildtype and mutant groups showed similar trends in the changes in expressions of the pluripotency markers and the genes involved in cardiac differentiation. For all groups, cardiac-troponin-T-positive cells were more than 80% at 14 days after differentiation (Appendix A). In addition, there were no significant differences in the cardiac troponinT–sarcomeric actinin network organization (Figure 1C) or size of CMs (Figure 1D) among *MeCP2^wildtype^*, *MeCP2^mutant^*, and control iPSC-CMs.

Although there were no differences in contractile function in terms of cell-shortening among the different groups of iPSC-CMs, a significantly higher incidence of irregular contractions was observed in the *MeCP2^mutant^* iPSC-CMs(31.97 ± 7.16%) compared with *MeCP2^wildtype^* (4.52 ± 2.50%) and control iPSC-CMs (2.50 ± 1.69%) (*p* < 0.01. Figure 2A). Moreover, the calcium (Ca2+)transient amplitude was significantly higher in the *MeCP2^mutant^* iPSC-CMs than in the *MeCP2^wildtype^* iPSC-CMs (28.04 ± 2.16 vs. 9.37 ± 0.75 normalized fluorescence unit, *p* < 0.05). Nevertheless, no significant differences in the Ca^2+^ transient maximal rising or recovery velocities were noted among the different groups of iPSC-CMs (Figure 2B). In addition, irregular peaks of calcium transients were sometimes observed in the *MeCP2^wildtype^* iPSC-CMs (Appendix A).

### 2.2. MeCP2-Mutation-Induced Abnormal Electrophysiological Properties of iPSC-CMs via Altered Expression of Multiple Cardiac Ion Channels

To evaluate the altered electrophysiological properties of *MeCP2^mutant^* iPSC-CMs, we recorded the field potentials and action potential durations of the different groups of iPSC-CMs using microelectrode array analysis and optical-mapping-based methods, respectively.

As shown in Figure 3A, the field potential duration recorded from the *MeCP2^mutant^* iPSC-CMs was significantly longer than that from the *MeCP2^wildtype^* iPSC-CMs (229.80 ± 17.92 ms vs. 135.50 ± 6.08 ms, *p* < 0.01), but there was no significant difference in the spontaneous beating rates. Similar to the microelectrode array analysis recording, optical mapping revealed the action potential duration (APD90) recorded from *MeCP2^mutant^* iPSC-CMs to be significantly longer than that recorded from the *MeCP2^wildtype^* iPSC-CMs (1007.00 ± 50.71 ms vs. 687.10 ± 24.75 ms, *p* < 0.01); again, there was no significant difference in the spontaneous beating rate (Figure 3B). The percentage of *MeCP2^mutant^* iPSC-CMs that exhibited early afterdepolarizations (EADs) was also significantly higher than that of the wildtype group(51.71 ± 33.99% vs. 11.82 ± 3.25%, *p* < 0.01), and the majority of EADs in the *MeCP2^mutant^* iPSC-CMs were observed in phase 2 (Figure 3B). On the other contrary, for those recording with no EADs, the APD90 value was similar between the *MeCP2^wildtype^* and *MeCP2^mutant^* iPSC-CMs (Appendix A).

To deduce the molecular mechanisms of these altered electrophysiological properties of *MeCP2^mutant^* iPSC-CMs, we determined the expression profile of 15 genes known to be associated with QT prolongation in different groups of iPSC-CMs [19]. As shown in Figure 4A, the expression of all those genes of interest, except *KCNJ5* and *SNTA1,* was significantly up-regulated in *MeCP2^mutant^* iPSC-CMs compared with that in *MeCP2^wildtype^* iPSC-CMs (all *p* < 0.05). Among these genes, substantial up-regulation of *CAV3* expression (64.01 ± 20.42-fold, *p* < 0.01) was observed in the *MeCP2^mutant^* iPSC-CMs and further confirmed by Western blot analysis. As shown in Figure 4B, the relative caveolin-3 protein level was significantly higher in *MeCP2^mutant^* iPSC-CMs compared with *MeCP2^wildtype^* iPSC-CMs (1.02 ± 0.02 vs. 0.60 ± 0.07, *p* < 0.05).

To evaluate the functional significance of this observation, we performed siRNA-mediated *CAV3* knockdown experiments in *MeCP2^mutant^* iPSC-CMs. As shown in Figure 4C, down-regulation of endogenous *CAV3* expression by approximately 70% using *CAV3*-specific siRNA (*p* < 0.05)did not significantly reduce the incidence of abnormal contraction or electrophysiological phenotypes observed in the control groups (transfected with scrambled siRNA). We speculated that the abnormal cellular electrophysiological properties with prolonged field potential duration and increased EADs in *MeCP2^mutant^* iPSC-CMs may not be a direct result of altered *CAV3* expression.

### 2.3. Calcium and Sodium Channel Blockers Ameliorate the Abnormal Cardiac Electrophysiological Properties Caused by MeCP2 Mutation

To further clarify the relative contribution of different ion channels to the prolonged field potential duration and increased incidence of EAD observed in *MeCP2^mutant^* iPSC-CMs, we treated them with different ion channel blockers.

As shown in Figure 5A, application of nifedipine (calcium channel blocker) at 0.5 µM significantly reduced the field potential duration from 222.70 ± 38.17 ms to 136.70 ± 16.71 ms (*p* < 0.01). In contrast, phenytoin (sodium channel blocker) and XE991 (potassium channel blocker) treatment at 0.5 µM showed no effect (not shown). When the concentration of these two blockers was increased to 5 µM, phenytoin significantly reduced the field potential duration from 207.00 ± 29.62 ms to 93.36 ± 2.66 ms (*p* < 0.01), while XE991 continued to show no significant effects. For the incidence of EAD, application of nifedipine but not phenytoin significantly reduced the incidence in *MeCP2^mutant^* iPSC-CMs from 54.43 ± 6.58% to 28.72 ± 8.58% (*p* < 0.05, Figure 5B).

Similar results were observed for the contraction profiles. As shown in Figure 5C,D, application of nifedipine, not phenytoin, reduced the frequency of irregular contractions in the *MeCP2^mutant^* iPSC-CMs from 31.97 ± 7.16% to 9.35 ± 5.96% (*p* < 0.05). These findings suggest that the increased incidence of EAD and irregular contractions could be due to excessive calcium influx, as the expressions of *CACNA1C* was significantly increased in the *MeCP2^mutant^* iPSC-CMs (Figure 4A).

### 2.4. Increased WNT Family Gene Expression Was Observed in MeCP2^mutant^ iPSC-CMs

Since MeCP2 protein is believed to play an important role in the regulation of gene transcriptions [9], we performed detailed transcriptome analysis using RNA sequencing analysis in different groups of iPSC-CMs (normal line IMR90 and *MeCP2^wildtype^* and *MeCP2^mutant^* iPSC-CMs). The full sets of raw and processed data can be accessed via the NCBI Gene Expression Omnibus database (accession number: GSE189983). Pathway analysis indicated that expression of numerous *WNT* family genes was altered in *MeCP2^mutant^* iPSC-CMs compared with *MeCP2^wildtype^* and control iPSC-CMs (Figure 6A).

On the basis of this observation, we used quantitative-PCR analysis to further evaluate the expression of 13 *WNT* genes that may affect cardiac functions. Among them, *WNT3A* (5.40 ± 2.24-fold, *p* < 0.05), *WNT5A* (3.09 ± 0.39-fold, *p* < 0.05), *WNT6* (11.53 ± 4.58-fold, *p* < 0.05), *WNT9A* (4.00 ± 1.67-fold, *p* < 0.05), and *WNT10A* (18.35 ± 9.62-fold, *p* < 0.05), were significantly up-regulated in *MeCP2^mutant^* iPSC-CMs compared with *MeCP2^wildtype^* iPSC-CMs (Figure 6B). On the basis of the putative role of MECP2 protein in altering CpG site methylation, we further performed genome-wide pyrosequencing methylation analysis in *MeCP2^mutant^* and *MeCP2^wildtype^* iPSCs-CMs. As shown in Figure 6C, differentially methylated CpG sites could be identified in *WNT1, WNT2, WNT2B, WNT 3A, WNT4, WNT5, WNT6, WNT7B, WNT8B, WNT9B*, and *WNT10A* from *MeCP2^mutant^* vs. *MeCP2^wildtype^* iPSCs-CMs. Indeed, except for the CpG sites of various WNT genes, reduced methylation was observed among different regions (especially the 5′UTR, CpG islands, and promoter regions) across the genome of *MeCP2^mutant^* iPSC-CMs, clearly indicating the role of MECP2 proteins in preservation of genomic methylation (Appendix A).

### 2.5. MECP2 Protein Regulates CACNA1C Expression via Canonical Wnt Signaling Pathway

The WNT signaling pathway has been implicated in the expression of L-type calcium channel *CACNA1C* [20]. To confirm the involvement of the Wnt signaling pathway in the altered *CACNAIC* expression observed in *MeCP2^mutant^* iPSCs-CMs, we further evaluated the β-catenin level in *MeCP2^mutant^* and *MeCP2^wildtype^* iPSC-CMs by Western blot analysis. As shown in Figure 7A, the relative β-catenin protein level was significantly higher in *MeCP2^mutant^* iPSC-CMs compared with *MeCP2^wildtype^* iPSCs-CMs(1.83 ± 0.26 vs. 0.42 ± 0.01, *p* < 0.05). Blocking of Wnt release by IWP2 did not alter the expression of *CACNAIC* in the *MeCP2^mutant^* iPSC-CMs (Figure 7B). On the contrary, inhibition of tankyrase by XAV939 significantly reduced *CACNAIC* expression (Figure 7B); it also decreased field potential duration from 331.30 ± 6.69 ms to 263.00 ± 2.76 ms (*p* < 0.05; Figure 7C) and the incidence of EAD in *MeCP2^mutant^* iPSC-CMs from 49.52 ± 4.65% to 12.64 ± 2.40% (*p* < 0.01; Figure 7D). This was likely due to the increased degradation of β-catenin protein upon XAV939 treatment (Appendix A).

To further confirm these observations, we evaluated the effects of *MeCP2* knockdown in control iPSC-CMs (normal line, female). As shown in Figure 7E, transfection of *MeCP2*-specific siRNA significantly reduced the relative expression of endogenous *MeCP2* (0.27 ± 0.05-fold, *p* < 0.05). Similar to *MeCP2^mutant^* iPSC-CMs, *MeCP2* knockdown in control iPSC-CMs (normal line) increased the expression of *CACNA1C* (6.43 ± 2.33-fold, *p* < 0.05), *WNT3A* (7.19 ± 1.43-fold, *p* < 0.05), *WNT4* (13.56 ± 2.74-fold, *p* < 0.05), *WNT6* (14.36 ± 6.78-fold, *p* < 0.05), and *CAV3* (28.21 ± 8.56-fold, *p* < 0.05) compared with the scrambled siRNA controls. Moreover, *MeCP2* knockdown in control iPSC-CMs significantly increased relative β-catenin protein level from 0.41 ± 0.03 to 0.79 ± 0.11 (*p* < 0.05; Figure 7F). Functionally, knockdown of endogenous *MeCP2* expression significantly increased the incidence of EAD from 8.33 ± 2.27% to 27.67 ± 4.83% (*p* < 0.01) and action potential duration from 947.50 ± 36.46 ms to 1176 ± 81.01 ms (*p* < 0.05; Figure 7G).

## 3. Discussion

In this study, we provide the first evidence using isogenic iPSCs derived from a heterozygous female RTT patient that a loss of function *MeCP2* mutation can directly alter the electrophysiological properties of human cardiomyocytes. Although there were no significant differences in the structure and morphology of CMs derived from wildtype and mutant RTT iPSCs, the *MeCP2^mutant^* iPSC-CMs exhibited significantly prolonged field potential duration and increased incidence of irregular contraction. These abnormal cellular electrophysiological properties of *MeCP2^mutant^* iPSC-CMs likely contribute to the occurrence of QT prolongation and sudden death observed in RTT patients.

Although loss of function MECP2 proteins are known to be the cause in the majority of RTT patients [8], the underlying mechanisms of the disease phenotype in different organs remain unclear. One challenge is that MECP2 protein affects the expression of thousands of genes after binding with methylated DNA; its effect is widespread and mostly subtle. It is also difficult to distinguish between changes in expression that are due to the primary effects of loss of MECP2 protein function and those that are due to secondary changes arising from dysfunction of different organs [21]. Previous studies [14] showed that nervous-system-specific conditional MeCP2 knock-out alone could induce secondary changes in the heart to cause QT prolongation. Nevertheless, this finding did not preclude any direct effects of loss of functional MECP2 protein on CMs. Furthermore, even within the same organ, the heterogeneity of different cell types as well as the mosaic pattern of *MeCP2* expression due to random X chromosome inactivation affected the results of transcriptomic profiling in RTT [21].

To address these issues, we performed functional analysis and transcriptomic profiling of a pair of isogenic iPSC-CMs derived from a patient with RTT. Our results demonstrated significantly up-regulated expression of multiple sodium, potassium, and calcium ion channels linked to long QT syndrome in *MeCP2^mutant^* iPSC-CMs. Although blockade of calcium and sodium channels using nifedipine and phenytoin, respectively, attenuated field potential prolongation in *MeCP2^mutant^* iPSC-CMs, only nifedipine significantly reduced the incidence of EADs. These observations suggest that changes in L-type calcium channel expression play a more important role in the electrophysiological abnormalities in *MeCP2^mutant^* iPSC-CMs.

Next, we further used transcriptomic analysis to evaluate whether *MeCP2* mutation directly contributed to the altered cardiac gene expression. Functionally, MECP2 protein may help to recruit co-repressor complexes to particular sequences of methylated DNA and modulate gene expression in a context-specific manner [22]. In such a case, the loss of functional MECP2 protein may have resulted in exposure of methylated CpG sites to DNA demethylases that rendered them less methylated and, in turn, facilitated the expression of the target genes. Using RNA sequencing analysis, we found that various *WNT* genes were up-regulated in *MeCP2^mutant^* iPSC-CMs. Genome-wide pyrosequencing methylation analysis further confirmed that a large number of CpG sites were less methylated in *MeCP2^mutant^* iPSC-CMs.

Altered activation of the Wnt signaling pathway has been implicated in the expression of L-type calcium channel *CACNA1C* [20]. In this study, we showed that increased expression of *WNT* genes in *MeCP2^mutant^* iPSC-CMs caused up-regulation of *CACNA1C* expression via the canonical β-catenin pathway. These findings were confirmed by knockdown of *MeCP2* expression using siRNA in normal iPSC-CMs. Moreover, inhibition of the Wnt/β-catenin pathway by XAV939 significantly reduced *CACNAIC* expression and decreased field potential duration of *MeCP2^mutant^* iPSC-CMs. This clearly demonstrated the contribution of Wnt/β-catenin signaling-pathway-dependent alteration of *CACNAIC* expression in the development of long QT phenotypes in RTT patients. It should be noted that in addition to *CACNA1C*, our result also revealed a marked increase in *CAV3* expression in *MeCP2^mutant^* iPSC-CMs. Nonetheless, suppression of *CAV3* expression in the *MeCP2^mutant^* iPSC-CMs did not significantly ameliorate the abnormal contraction or electrophysiological conditions due to *MeCP2* mutation. This was likely because, unlike most long-QT-related genes, *CAV3* encodes caveolin-3 protein that is not directly involved in ion channel formation but rather functions to modulate activity of different non-related ion channels [23]. As well as stimulating calcium channel opening [24], caveolin-3 negatively regulates potassium channel activity [25]. In addition, since sodium and potassium currents are usually in opposite directions [26], the effects of increased potassium channel expression may be reduced by the increased expression of different sodium channel genes. This may explain why inhibition of excessive calcium channel activity appears to be more effective in correcting the cardiac abnormalities observed in *MeCP2^muatant^* iPSCs-CMs, even though we also observed significant up-regulation of different potassium channel related genes in the same cell line.

Our study has limitations. Our iPSC isogenic cell lines were generated from somatic cell samples obtained from an anonymous 3-year-old female patient with unclear cardiac phenotype. As a result, we were not able to compare the cardiac phenotype, including QT interval on electrocardiogram, with our iPSC-CM model. The potential pathophysiological roles of Wnt/β-catenin signaling in RTT need to be confirmed in future studies of RTT patients with QT prolongation.

## 4. Materials and Methods

### 4.1. RTT Patient-Specific Isogenic iPSCs

Isogenic iPSC lines were obtained from Colman A et al. [18]. In brief, the female patient carried a heterozygous mutation with a 32 bp deletion within the 3′ coding region of the *MeCP2* gene (*MeCP2-1155del32*) [18]. In this report, the isogenic iPSC lines that expressed only the specific *MeCP2* allele (either wildtype or mutant) were denoted as *MeCP2^wildtype^* or *MeCP2^mutant^*, respectively. For normal controls, a non-disease iPSC line, IMR90 (female, WiCell, Madison, WI, USA) was used.

### 4.2. Maintenance and Cardiac Differentiation of iPSCs

The iPSC lines were maintained in a serum-free and feeder-free system, as previously described [27,28].Cardiac differentiation was instigated using the PSC Cardiomyocyte Differentiation Kit (Gibco, Waltham, MA, USA) according to the manufacturer’s instructions. In brief, undifferentiated iPSCs were seeded on 12-well plates coated with GelTrex (Gibco, Waltham, MA, USA) and cultured in StemFlex (Gibco, Waltham, MA, USA) media to 90–100% confluence. Afterward, the cells were cultured subsequently in medium A (first 2 days), medium B (next 2 days), and cardiomyocyte maintenance medium (for approximately 7–14 days). Beating human iPSC-CMs were dissociated with collagenase B and subjected to immunostaining using antibodies specific for cardiac troponin T or sarcomeric α-actinin [28]. To evaluate specific functional properties of interest, iPSC-CMs were treated with different drugs or transfected with siRNAs specific to *CAV3* (S2454, Ambion, Waltham, MA, USA) or *MeCP2* (AM16708, Ambion, Waltham, MA, USA).

### 4.3. Genomic DNA and mRNA Analysis

Genomic DNA and total RNA of iPSC-CMs were extracted using the PureLink Genomic DNA Mini Kit (Invitrogen, Waltham, MA, USA) and RNAqueous Total RNA Isolation Kit (Ambion, Waltham, MA, USA) Kit, respectively. For PCR-based analysis, cDNAs were prepared from total RNA using the QuantiTect Reverse Transcription Kit (Qiagen, Hilden, Germany). The presence of specific *MeCP2* allele and its expression (wildtype and mutants) was determined using the PCR-based method described previously [18].

### 4.4. Field Potential Recordings

Field potential recordings were obtained using a microelectrode array system (MEA) (Multi Channel Systems, Reutlingen, Germany). In brief, iPSC-CMs were mechanically dissociated into small clusters (approximately 10–30 cells) and seeded onto a MEAs chamber with 60 electrodes (MEAs with 60 electrodes) for 1 day prior to the experiment. Data acquisition and analysis were performed using Cardio2D Software (Version 1.0.0) (Multi Channel Systems, Reutlingen, Germany).

### 4.5. Fluorescent-Potentiometric-Probe-Based Membrane Potential Evaluation

Screening of action potential alteration was performed using the optical-mapping-based method [29]. In brief, iPSCs-CMs were enzymatically digested (with 1 mg/mL collagenase) into single cells and seeded onto gelatin-coated glass coverslips. The dissociated iPSC-CMs were cultured for 3 days prior to membrane potential evaluation. Prior to recording, cells of interest in the monolayer were loaded with 2 µM of Di-8-ANEPPS (Invitrogen) for 15 min. Unbound dyes were removed, and the cells were transferred to a temperature-controlled chamber (37 °C) and studied with an inverted fluorescent microscope (Olympus IX51, Olympus, Tokyo, Japan). In the absence of pacing, cells were illuminated with a monochromic light source at 480 nm and emission at 635 nm captured by the MiCamUltima (Scimedia, Costa Mesa, CA, USA) ultra-fast CMOS sensor (objective: 20×; resolution: 128 pixels × 128 pixels). Recordings were performed at 1–2 msec/frame using Ultima Acquisition software (Ver.1111) (Scimedia, Costa Mesa, CA, USA). Experimental data were analysed using BV_Ana software (Ver.1312) (Scimedia, Costa Mesa, CA, USA). EAD was defined as a positive fluorescent signal deflection of >5% of the amplitude of the action potential transient following the initial repolarization (as indicated by the first decrease in fluorescent signal), and the incidence of EAD was calculated from 10 recordings of 5 consecutive action potential transients. 

### 4.6. IonOptix Cellular Contractility and Calcium Transient Analysis

The contraction properties of iPSC-CMs were evaluated using the IonOptix contractility system (IonOptix, Milton, MA, USA). In brief, iPSC-CM was placed in the experimental chamber (individually or in small clusters) and paced with a MyoPacer stimulator (10 V, 1–4 Hz). Change in cell length was recorded by video-based edge detection with the MyoCam-S camera attached to an inverted microscope (Olympus, IX-51, Olympus, Tokyo, Japan). In brief, the left and right edges (appearing as the darkest lines) of a cell of interest were defined manually and marked as the reference points for automated tracing using the softedge acquisition add-on (IonOtix, Milton, MA, USA). The cell-shortening, as indicated by the changes in the distances between the reference points, were automatically recorded at 240 frames/s. For calcium transient analysis, iPSC-CMs were incubated with 5μM fura-2 AM (ThermoFisher scientific, Waltham, MA, USA) for 20 min, unincorporated dye was removed, and the change in fluorescence was recorded with a Myocam-S camera. Data acquisition and analysis were performed using IonWizard 6.3 software (IonOptix, Milton, MA, USA).

### 4.7. Quantitative PCR Analysis

Relative expression of the genes of interest was evaluated using real-time PCR analysis. In brief, genes of interest were amplified using PowerUp SYBR Green Master Mix (Applied Biosystems, Waltham, MA, USA) with gene-specific primers (Appendix A). Amplification was performed using the StepOnePlus Real-Time PCR System (Applied Biosystems, Waltham, MA, USA). The relative gene expression was determined with the comparative Ct method, where the expression of *TNNT2* or *GAPDH* were used as the internal references.

### 4.8. Transcriptome Analysis

RNA sequencing was performed to compare the difference in transcription profile among different cell lines of interest. In brief, total RNA was isolated using the mirVana miRNA Isolation Kit (Ambion, Waltham, MA, USA). Ribosomal RNA (rRNA) depletion was performed using aNEBNext^®^ rRNA Depletion Kit (Human/Mouse/Rat) (NEB, Ipswich, MA, USA). cDNA libraries were prepared using aNEBNext^®^ Ultra II Directional RNA Library Prep Kit (NEB). The cDNA libraries were submitted to the Centre for Panoromic Sciences (The University of Hong Kong) for Solexa sequencing. For quality control, sequencing reads were first filtered for adapter sequence and low-quality sequence, followed by retaining only reads with read length ≥ 40 bp. Low quality was defined as: (1) reads with more than 5% unknown bases (“N”) or (2) reads having more than 50% of bases with quality value ≤ 11. Subsequently, sequencing reads were filtered for rRNA sequence, and remaining reads were used for downstream analysis. Filtered reads were aligned to reference Human Genome (GRCh38). Raw data were analyzed by Illumina Real Time Analysis. The differential expression analysis was performed using Baggerley’s test and the methods described previously [30]. Pathway analysis was conducted using the Partek Genomics Suite with Partek Pathway (Partek, Chesterfield, MO, USA), in which biological interpretation was analysed with the pathway enrichment with KEGG database (Kanehisa Laboratories, Kyoto, Japan) for human. 

### 4.9. Genomic DNA Methylation Analysis

Genomic DNA methylation analysis was performed as described in our previous publication [28]. For data analysis, Reduced Representation Bisufite-Seq (RRBS) data were used for alignment, quality control, and methylation calling, as described previously [31,32]. In brief, to improve mapping efficiency and accuracy, adapter- and quality-trimming were first performed by Trim Galore (https://github.com/FelixKrueger/TrimGalore) (accessed on 4 August 2020) on the raw sequencing reads of the two samples to remove Illumina adapter sequences, low-quality bases (base quality < 25) and, in particular, artificially introduced cytosine during library preparation using the -RRBS mode. The trimmed sequencing reads were then aligned to the reference (human genome assembly hg19) by bwa-meth (https://github.com/brentp/bwa-meth) (accessed on 4 August 2020) [33]. Then, MethylDackel (https://github.com/dpryan79/MethylDackel) (accessed on 4 August 2020) was used to extract the methylation calls from the aligned bam files and the perCpG matrix for those CpG sites with a minimum coverage of 8 was calculated. For each CpG site, the number of reads supporting the C allele (methylated) and the number of reads supporting the T allele (unmethylated) were recorded. Methylation level for each CpG was then calculated as the number of reads supporting the C allele divided by the total number of reads covering that site. HOMER [34] was used with the assign Genome Annotation tool to annotate the genomic features of TSS (transcription start site), TTS (transcription termination site), exon (coding), 5′ UTR exon, 3′ UTR exon, intronic, or intergenic for the given CpGs. CpG shore was defined as the 2 kb region flanking the CpG islands. Coverage and methylation level of the CpG site in each genomic feature (TSS, CpG island, and CpG shore) were plotted and compared with published data to ensure data quality. Finally, the CpG sites located in 13 genes (*WNT1*, *WNT2*, *WNT2B*, *WNT3A, WNT4*, *WNT5A*, *WNT6*, *WNT7B*, *WNT8A*, *WNT8B*, *WNT9A*, *WNT9B*, and *WNT10A*) were extracted to compare the difference in methylation level between two samples.

### 4.10. Western Blot Analysis

After harvest, the cells of interest were lysed in RIPA buffer (abcam, Boston, MA, USA) supplemented with protease inhibitor cocktail (P8340-sigma-Aldrich, Darmstadt, Germany). Insoluble debris was removed by centrifugation (10,000× *g*, 15 min, at 4 °C). Protein concentration was determined using Pierce BCA Protein Assay Kit, following the instruction provided by the manufacturer (ThermoFisher, Waltham, MA, USA). Protein samples (20 µg) were resolved on 10% polyacrylamide gels and electrically transferred onto PVDF membranes (ThermoFisher, Waltham, MA, USA). Following blocking with superblock-T20 (ThermoFisher, Waltham, MA, USA), the membranes were incubated with primary antibodies of interest and detected using species-match HRP-conjugated secondary antibodies (ThermoFisher, Waltham, MA, USA). Target proteins were visualized by enhanced chemiluminescence (ECL)-based method using Pierce ECL Plus Western Blotting Substrate (ThermoFisher, Waltham, MA, USA). ECL signals were acquired using Chemidoc imaging system (Bio-Rad, Hercules, CA, USA) and quantified using the Image J software (Version 1.53a) (National Institute of Health, Bethesda, MD, USA).

### 4.11. Statistical Analysis

All data are expressed as mean ± SEM from at least 3 sets of independent experiments on at least 3 sets of biological replicates unless otherwise specified. Statistical analyses were performed using Prizm software (Version 5.00) (GraphPad Inc., San Diego, CA, USA). Comparison of parameters among different groups was performed using the Mann–Whitney U test or one-way analysis of variance with Tukey’s test, as appropriate. Differences were considered statistically significant at a level of *p* < 0.05.

## 5. Conclusions

In conclusion, our study provides important insight into the involvement of altered Wnt/β-catenin signaling and ion channel imbalance in the intrinsic cardiac dysfunction associated with *MeCP2 mutation*. This may help reveal the etiology that contributes to the development of cardiac manifestations in RTT patients.

## Figures and Tables

**Figure 1 ijms-23-15609-f001:**
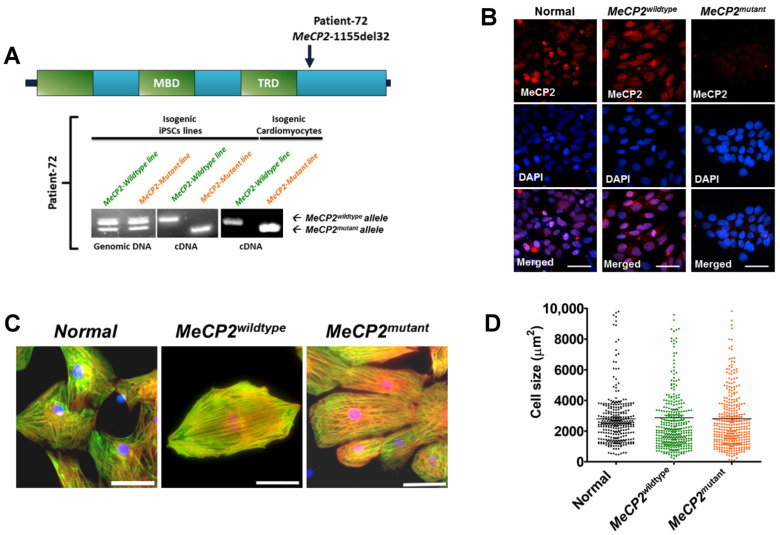
Generation of isogenic iPSC-derived cardiomyocytes expressing different *MeCP2* alleles. (**A**) The *MeCP2* mutation is indicated by the arrow head, and the expression of specific *MeCP2* allele in the isogenic iPSCs was confirmed by PCR analysis. (**B**) Immunostaining analysis of the isogenic iPSCs with MECP2-specific antibody (ThermoFisher, Waltham, MA, USA, MA5-33096) indicated the nucleic localization of wildtype MECP2 protein. (**C**) The iPSC lines were differentiated into cardiomyocytes (iPSC-CMs) and co-stained with cardiac troponin T (green, Abcam, Cambridge, UK, ab45932) and sarcomeric α-actinin (red, sigma, A7811), and the nuclei were stained with DAPI (blue). (**D**) The cell size of the iPSC-CMs. N = 284, 357, and 360 cells measured from normal, *MeCP2^wildtype^*, and *MeCP2^mutant^* groups, respectively. Data are represented as mean ± SEM. Scale bar: 100 µm. (See also Appendix A).

**Figure 2 ijms-23-15609-f002:**
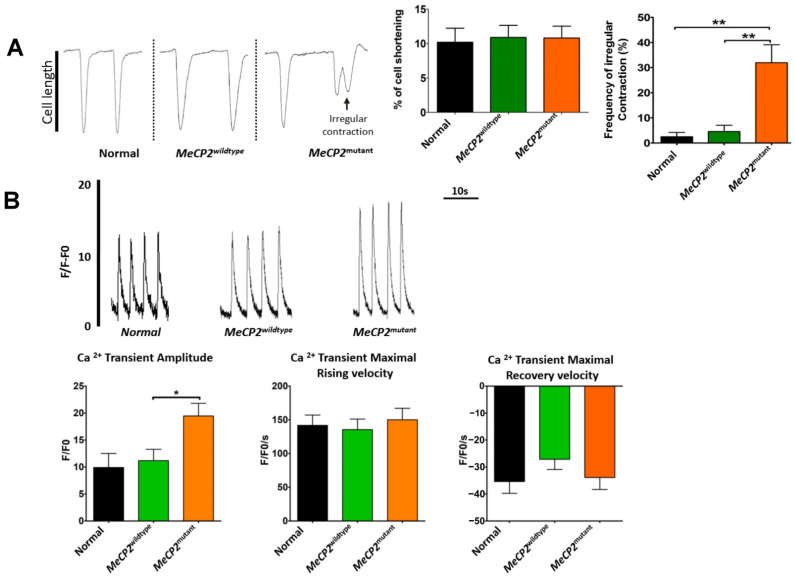
*MeCP2^muatant^* iPSC-derived cardiomyocytes showed higher frequency of irregular contraction but no difference in calcium handling properties. (**A**) The changes in cell length of cardiomyocytes derived from normal (IMR90) and Rett syndrome patient’s isogenic iPSCs were measured with the IonOptix contractility system. Left panel: representative tracings. Middle and right panels: statistical analysis of the % cell-shortening (N = individual recordings of 20 cells from the normal groups and 25 cells from each of the *MeCP2^wildtype^* and *MeCP2^mutant^* groups) and the frequency of irregular contractions (N = 10 calculations of 10 sets of data, with each data set containing 5–10 independent recordings). (**B**) Intracellular calcium transients recorded with the IonOptix calcium system. Upper panel: representative tracings of intracellular calcium (Ca^2+^) transients. Lower panels: quantitative analysis of the Ca^2+^ transient amplitudes, maximal rising, and recovery velocities. N = 20 individual recordings of 20 cells from each group. Data are represented as mean ± SEM. * *p* < 0.05, ** *p* < 0.01. (See also Appendix A).

**Figure 3 ijms-23-15609-f003:**
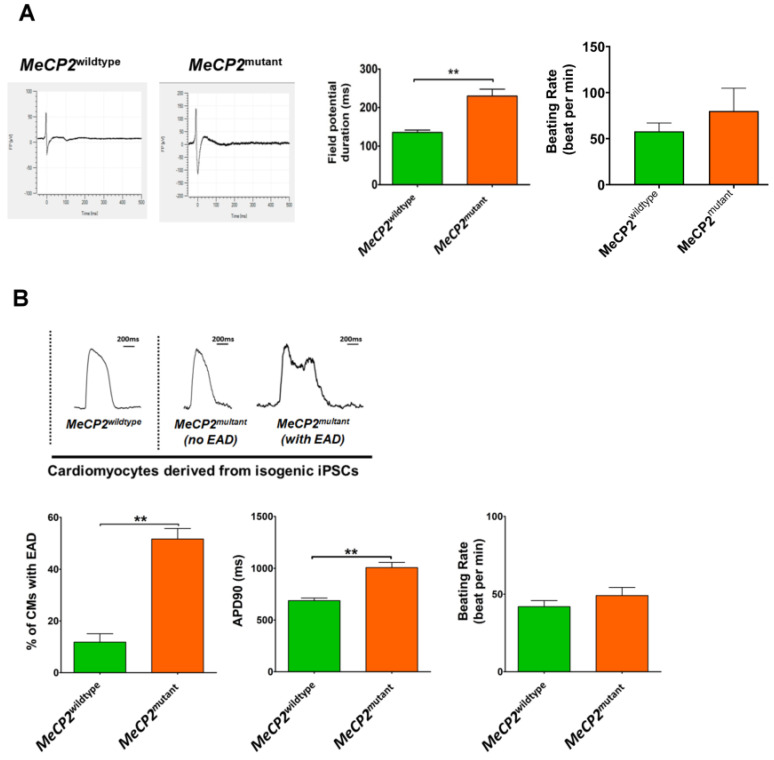
Cardiomyocytes expressing *MeCP2^mutant^* exhibited abnormal electrophysiological properties. (**A**) Representative tracings (left panel), quantitative analysis of the extracellular field potentials recorded by multi-electrode array system (MEA) (middle panel), and the spontaneous beating rates (right panel) of the iPSC-CMs. N = 10 individual recordings from 10 electrodes from 2 MEA plates for each group. (**B**) Representative tracing (upper panel), quantitative analysis of incidences of EAD (lower left middle panel), membrane potential durations (APD90) measured with optical mapping system (lower middle panel), and spontaneous beating rates (lower right panel) of the iPSC-CMs. For the calculation of the percentage of the incidence of EAD, N = 10 calculations of 10 collections of data sets, with each data set comprising 5–10 independent recordings. For APD90 evaluation, N = 50 individual recordings from 50 small cell clusters (2–3 cells/cluster) for each group. Data are represented as mean ± SEM. ** *p* < 0.01. (See also Appendix A).

**Figure 4 ijms-23-15609-f004:**
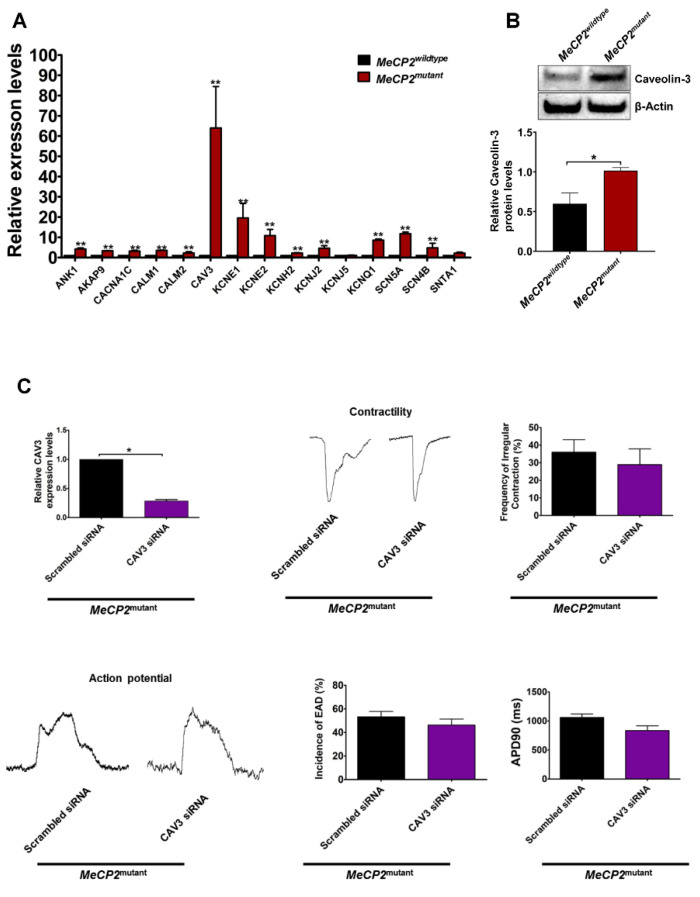
Quantitative PCR analysis of long-QT-syndrome-associated genes. (**A**) Total RNA was extracted from iPSC-derived cardiomyocytes and reverse transcribed into cDNAs. The cDNAs were used as templates for quantitative PCR analysis. N = 4 individual experiments on 4 independent samples collected from each group. (**B**) Western blot analysis of the caveolin-3 protein levels (Caveolin-3-specific antibody: Santa Cruz, sc-5310; β-Actin-specific antibody: Santa Cruz, sc-8432). N = 3 individual experiments on 3 independent samples collected from each group. (**C**) *MeCP2^mutant^* iPSC-CMs were transfected with *CAV3*-specific siRNA or scrambled siRNA (as control), and the following changes were calculated: *CAV3* expression (upper left panel, N = 4 individual experiments on 4 independent samples collected from each group), contractile properties (upper center and right panels, respectively, N = 10 individual calculations of the 50 individual recordings from each group), and electrophysiological properties (lower panels, N = 10 individual recordings from 10 independent samples prepared from each group). Data are represented as mean ± SEM. * *p* < 0.05, ** *p* < 0.01, compared with *MeCP2^wildtype^* group for Figure 4A or the indicated groups for other sections.

**Figure 5 ijms-23-15609-f005:**
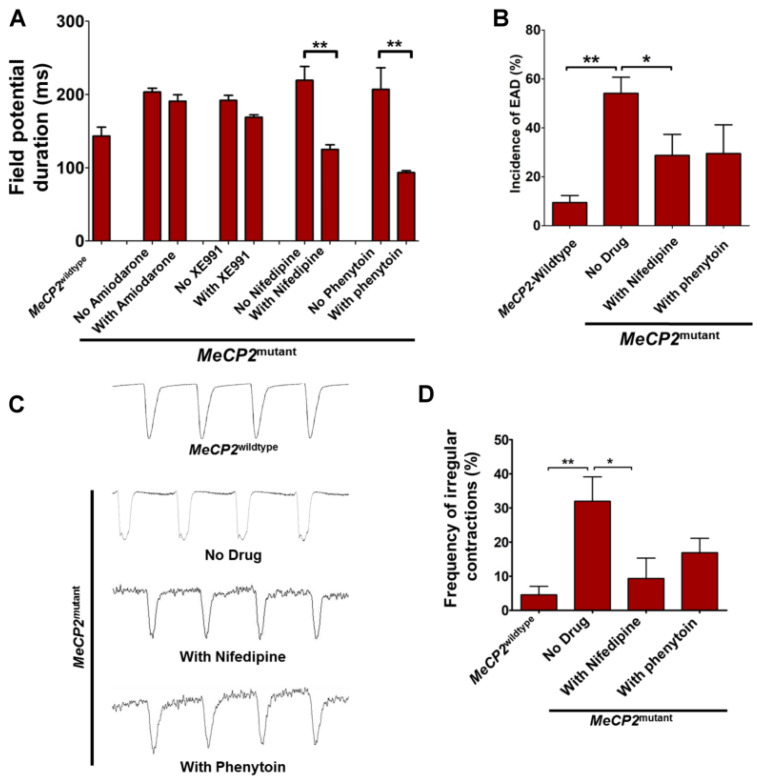
Calcium and sodium channel blockers ameliorate the abnormal cardiac phenotypes in cardiomyocytes expressing the *MeCP2^mutant^* allele. (**A**) iPSC-CMs were treated with 1 µM amiodarone (anti-arrhythmic drug), 5 µM XE991 (KCNQ channel blocker channel blocker), 5 µM phenytoin (voltage-gated sodium channel blocker), or 0.5 µM nifedipine (L-type calcium channel blocker). The field potential duration was measured with multi-electrode array (MEA) analysis before and 1 min after the addition of drugs. N = 5 individual measurements from 5 recordings from 2 MEA plates from each group. (**B**) iPSC-CMs were treated with 0.5 µM nifedipine and 5 µM phenytoin, and changes in the incidence of EAD were evaluated using an optical-mapping-based method. N = 10 calculations of 10 collections of data sets, with each data set containing 5–10 independent recordings. (**C**) iPSC-CMs were treated with 0.5 µM nifedipine or 5 µM phenytoin, and changes in incidence of irregular contractions were evaluated using the IonOptix contractility system. (**D**) Quantitative analysis of the frequency of irregular contractions. N = 7 calculations of the 7 data sets (each data sets contained 5–10 independent recordings) collected from each group. Data are represented as mean ± SEM. * *p* < 0.05, ** *p* < 0.01.

**Figure 6 ijms-23-15609-f006:**
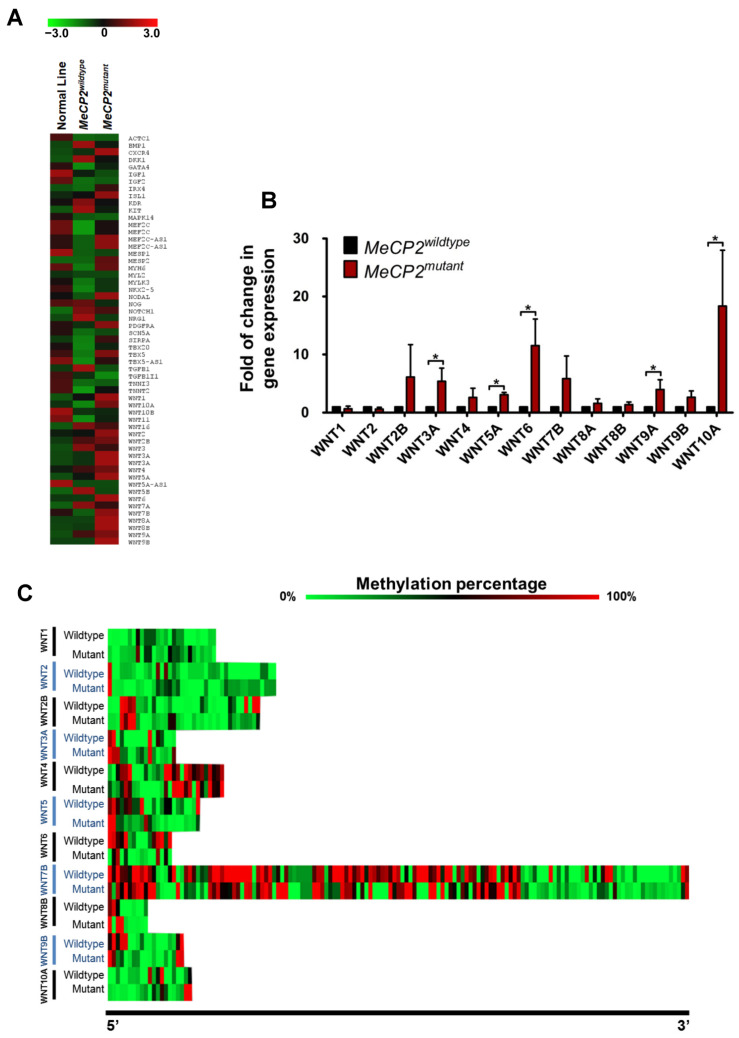
Up-regulated expression of *WNT* genes is associated with *MeCP2* mutation. (**A**) Heat map analysis of the expression of *WNT* genes as identified by RNA-sequencing analysis in the iPSC-derived cardiomyocytes. (**B**) Quantitative PCR analysis of WNT family members in cardiomyocytes derived from isogenic iPSC lines from a Rett syndrome patient. N = 4 individual experiments from 4 independent samples collected from each group. (**C**) Heatmap analysis of the methylation status of selected WNT family members. Data are represented as mean ± SEM. * *p* < 0.05. (See also Appendix A).

**Figure 7 ijms-23-15609-f007:**
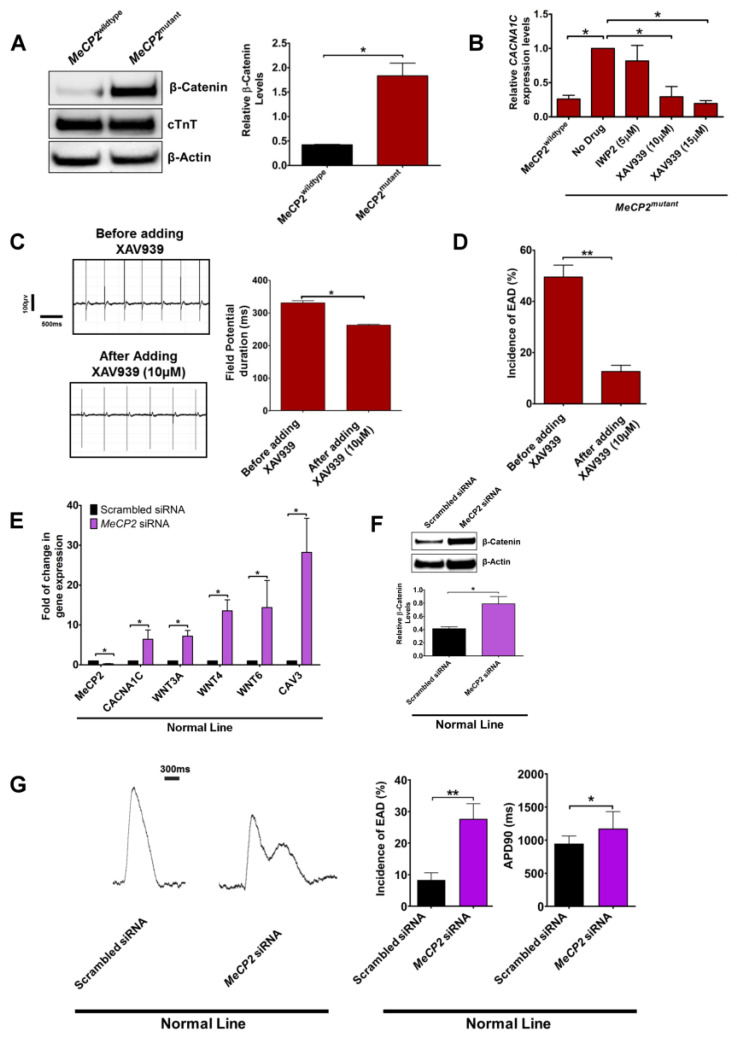
*MeCP2* mutation up-regulated *CACNAIC* expression via activation of the canonical Wnt signaling pathway. (**A**) Western blot analysis of β-catenin protein levels (β-catenin-specific antibody: ThermoFisher, MA5-34961). N = 3 individual experiments from 3 independent samples collected from each group. (**B**) Effects of Wnt signaling inhibitors on *CACNAIC* expression in the isogenic iPSC-CMs lines. N = 4 individual experiments from 4 independent samples collected from each group. (**C**) Effects of XAV939 on field potential duration recorded from *MeCP2^mutant^* iPSC-CMs. N = 3 recordings from 3 independent samples seeded on 3 different MEA plates. (**D**) Effects of XAV939 on the incidence of EAD in *MeCP2^mutant^* iPSC-CMs (left panel), a representative tracing of action potential showing EAD remained after XAV99 (right panel). N = 20 calculations of 20 collected data sets, with each data set comprising 5–10 independent recordings. (**E**) Effects of *MeCP2* knockdown on the expression of *CACNA1C, WNT3A, WNT4, WNT6, and CAV*. N = 3 individual experiments from 3 independent samples collected from each group. (**F**) Western blot analysis of β-catenin protein level in iPSC-CMs with *MeCP2* knockdown. N = 3 individual experiments from 3 independent samples collected from each group. (**G**) Effects of *MeCP2* knockdown on the membrane potential. Left panel: representative tracings. Right panel: incidences of EAD (N = 8 calculations of 8 data sets, with each data set comprising 5–10 independent recordings) and APD values (N = 10 individual recordings from 10 clusters of cells from each group) of the iPSC-CMs transfected with scrambled siRNA or *MeCP2*-specific siRNA. Data are represented as mean ± SEM. * *p* < 0.05, ** *p* < 0.01. (See also Appendix A).

## Data Availability

The full sets of raw and process data can be accessed freely via the NCBI Gene Expression Omnibus database (accession number: GSE189983) upon the publication of this manuscript.

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
