# Peer review of "Isogenic Human-Induced Pluripotent Stem-Cell-Derived Cardiomyocytes Reveal Activation of Wnt Signaling Pathways Underlying Intrinsic Cardiac Abnormalities in Rett Syndrome"

_ijms, 2022, doi:10.3390/ijms232415609_

Round 1

Reviewer 1 Report

First of all, I would like to congratulate the authors for their work. I have only a few minor considerations: 

1) In the introduction section: I personally would eliminate the sentence that advances us the results, and that begins with "Based on detailed functional and transcriptomic analysis...". 

2) The profile/phenotype of the patient chosen is also a limitation, but it is already referred to by the authors, so I do not see how it can be remedied and recognizing the limitation I think the problem is partially solved. 

3) In conclusions: not adding the hypothesis of alteration of ionic expression in neurons may be the reason for neurological dysfunction. It is something that has not been studied. It could make sense instead to add information in this sense and in the discussion and to compare with other similar clinical pictures in which this dysfunction has already been described. 

Author Response

Reviewer 1

Comments and Suggestions for Authors

Comment 1:

First of all, I would like to congratulate the authors for their work. I have only a few minor considerations: 

In the introduction section: I personally would eliminate the sentence that advances us the results, and that begins with "Based on detailed functional and transcriptomic analysis...". 

Response to comment 1:

We thank the reviewer for appreciating our efforts and works. As suggested, we have rephrased the statement “ Based on detailed… “ into “ using “Using these isogenic MeCP2wildtype and MeCP2mutant iPSC-CMs as platform”. For detail, please refer to introduction section the revised manuscript (Page 4, line 107)

Comment 2:

2) The profile/phenotype of the patient chosen is also a limitation, but it is already referred to by the authors, so I do not see how it can be remedied and recognizing the limitation I think the problem is partially solved. 

Response to comment 2:

We thank the reviewer for understanding our limitation. Addressing this issue, we have included the MeCP2 knock-down experiment to confirm the observations, and we hope this could help the readers to understand the mechanism that we suggested in our study.    

Comment 3:

In conclusions: not adding the hypothesis of alteration of ionic expression in neurons may be the reason for neurological dysfunction. It is something that has not been studied. It could make sense instead to add information in this sense and in the discussion and to compare with other similar clinical pictures in which this dysfunction has already been described. 

Response to comment 3

As suggested, we have removed the sentences regarding the neuronal dysfunctions in the conclusion section, and adding a brief statement to outline the contribution of the MeCP2 mutations in the development of cardiac phenotypes in RTT patient. For detail, please refer to the conclusion section of the revised manuscript (Page 14, line 457) 

Reviewer 2 Report

Journal IJMS (ISSN 1422-0067)

Manuscript ID ijms-2013609

Type Article

Title

Isogenic Human-Induced Pluripotent Stem Cell-Derived Cardiomyocytes Reveal Activation of Wnt Signaling Pathways Underlying intrinsic cardiac abnormalities in Rett Syndrome

Authors

Kwong-Man Ng, Qianqian Ding, Yiu-Lam Tse, Oscar Hou-In Chou, Wing-Hon 1Wing-Hon Lai, Ka-Wing Au, Yee-Man Lau, Yue Ji, Chung-Wah Siu, ClaraSze-Man Tang, Alan Colman, Suk-Ying Tsang, Hung-Fat Tse.

General Comment:

The findings of this study performed in vitro support the idea that the Wnt signaling pathways are involved in intrinsic cardiac abnormalities in Rett Syndrome (RTT) by using isogenic iPSC-derived cardiomyocytes that expressed either MeCP2 wildtype or MeCP2 mutant allele, and iPSC-CMs from a non-affected female control. Methyl-CpG-binding protein 2 (MeCP2) is an X-linked epigenetic modulator whose dosage is critical for neural development and function and it is well known that loss-of-function mutations in MECP2 cause RTT.

The main objective of this study is to identify/confirm signaling pathways and potential quantitative biomarkers that could aid early diagnosis and/or the monitoring of disease progression. Functional studies, RNA-seq, Wnt signaling analysis (inhibitor treatment and siRNA-based gene silencing) identified abnormal electrophysiological properties, prolonged action potential and increased frequency of spontaneous early after polarization, up-regulation of various Wnt family genes in MeCP2 mutant iPSC-CMs. Also, treatment of MeCP2 mutant iPSC-CMs with a WNT inhibitor XAV939 significantly decreased β-catenin protein level and CACN1AC expression and ameliorated their abnormal electrophysiological properties. The data provide novel insight into the activation of the altered Wnt-β-catenin signaling pathway and ion channel imbalance that underlies the intrinsic cardiac abnormalities associated with MeCP2 mutations in RTT.

Overall, this manuscript is well written and up-to-date.

Apart from all these things in this manuscript, I recommend publication after addressing all my queries. Moreover, I also suggest that the authors thoroughly revise the manuscript for grammatical errors and syntactic.

My main criticisms relate to the following issues:

Major issues

First of all, I would like to appreciate the authors of this work, however Authors should consider to revise the manuscript and add additional data to improve the manuscript for publication.

For my detailed comments, I request Authors to see my comments below in a pointwise manner.

1)    I recommend the Authors of this work to provide in details the cardiac differentiation protocol in materials and methods for example a time line, if small molecule were used for the differentiation protocol, if the hiPS were propagated in the undifferentiated state on top of the feeder layer, if any coating was used.

2)    The Authors should provide the evaluation of the temporal gene expression pattern associated with this in vitro differentiation process such as: initial decrease in the expression of undifferentiated pluripotent markers (OCT4 and NANOG) at day x, cardio-mesoderm markers (Brachyury and MESP1) at x days of differentiation. This should be followed by expression of cardiac progenitor markers (Isl-1, a marker of secondary heart field progenitors) and cardiac-associated transcription factors (Nkx2.5, Mef-2c, and Gata-4) and last markers of cardiac-specific structural genes, including sarcomeric-related proteins (α-myosin heavy chain [MYH6], β-myosin heavy chain [MYH7], and cardiac troponin-I [CTNNI]) and ion channel proteins. I am suggesting some, but the Authors can provide other markers if they have already analyzed them.

3)    The Authors should provide in material and methods QC analysis of RNA and DNA, how Pathway analysis was realized which algorithm has been used for that; and clarify how contractile function analysis in terms of cell shortening was evaluated (describe in detail how videos were analyzed).

4)    I recommend the Authors to describe western blotting analysis, what program did you use to quantize the bands? It does not appear in the manuscript.

5)    Finally, in addition to the above points, consider re-writing the conclusion part along with prospects. In this, the Authors should also discuss the real-life utility of this work and the author's future aims and objectives with these as-prepared materials as it should also be mentioned because science should be done to get real outcomes, not just publication-based outcomes. I believe that this is intended to improve the quality of the manuscript that is deemed suitable for publication.

Minor issues

English needs to be improved, there are some words and expressions that need to be revised, for example add a single space after a period or any other punctuation mark you use to end a sentence in the line 361 (iPSC-CMscaused), line 401 ([28].To).

Author Response

Reviewer 2

General Comment:

The findings of this study performed in vitro support the idea that the Wnt signaling pathways are involved in intrinsic cardiac abnormalities in Rett Syndrome (RTT) by using isogenic iPSC-derived cardiomyocytes that expressed either MeCP2 wildtype or MeCP2 mutant allele, and iPSC-CMs from a non-affected female control. Methyl-CpG-binding protein 2 (MeCP2) is an X-linked epigenetic modulator whose dosage is critical for neural development and function and it is well known that loss-of-function mutations in MECP2 cause RTT.

The main objective of this study is to identify/confirm signaling pathways and potential quantitative biomarkers that could aid early diagnosis and/or the monitoring of disease progression. Functional studies, RNA-seq, Wnt signaling analysis (inhibitor treatment and siRNA-based gene silencing) identified abnormal electrophysiological properties, prolonged action potential and increased frequency of spontaneous early after polarization, up-regulation of various Wnt family genes in MeCP2 mutant iPSC-CMs. Also, treatment of MeCP2 mutant iPSC-CMs with a WNT inhibitor XAV939 significantly decreased β-catenin protein level and CACN1AC expression and ameliorated their abnormal electrophysiological properties. The data provide novel insight into the activation of the altered Wnt-β-catenin signaling pathway and ion channel imbalance that underlies the intrinsic cardiac abnormalities associated with MeCP2 mutations in RTT.

Overall, this manuscript is well written and up-to-date.

Apart from all these things in this manuscript, I recommend publication after addressing all my queries. Moreover, I also suggest that the authors thoroughly revise the manuscript for grammatical errors and syntactic.

My main criticisms relate to the following issues:

Major issues

First of all, I would like to appreciate the authors of this work, however Authors should consider to revise the manuscript and add additional data to improve the manuscript for publication.

For my detailed comments, I request Authors to see my comments below in a pointwise manner.

Comments 1:

I recommend the Authors of this work to provide in details the cardiac differentiation protocol in materials and methods for example a time line, if small molecule were used for the differentiation protocol, if the hiPS were propagated in the undifferentiated state on top of the feeder layer, if any coating was used.

Response to comment 1:

We thank the reviewer for the valuable comments. For the cardiac differentiation, we used the PSC Cardiomyocyte Differentiation Kit (A2921201) from Gibco. We checked again with the user manual and product notes provided by manufacturer, yet, no information on the small molecules within the medium were provided. (This may be due to  commercial reason). For the differentiation, monolayer of iPSCs were cultured on feeder-free condition in 12 well plates-coated with Geltrex (A1413201  Gibco). We have added the details in the “materials and methods” section of the revised manuscript. Please refer to Page 10, line 323 for detail.    

Comment 2:

The Authors should provide the evaluation of the temporal gene expression pattern associated with this in vitro differentiation process such as: initial decrease in the expression of undifferentiated pluripotent markers (OCT4 and NANOG) at day x, cardio-mesoderm markers (Brachyury and MESP1) at x days of differentiation. This should be followed by expression of cardiac progenitor markers (Isl-1, a marker of secondary heart field progenitors) and cardiac-associated transcription factors (Nkx2.5, Mef-2c, and Gata-4) and last markers of cardiac-specific structural genes, including sarcomeric-related proteins (α-myosin heavy chain [MYH6], β-myosin heavy chain [MYH7], and cardiac troponin-I [CTNNI]) and ion channel proteins. I am suggesting some, but the Authors can provide other markers if they have already analyzed them.

Response to comment 2:

We thank the reviewer for the comment. We also think this is a good suggestion to look at the changes in the expressions of the genes listed by the reviewer during the cardiac differentiation, especially because the manufacturer did not provide the detailed component of the differentiation medium. As suggested, we have collected cell samples at different stages and performed real-time quantitative PCR analysis of the gene listed. The results were showed in the revised Figure S1. However, as the results obtained from the cTNNI was not valid ( likely due to non-specific priming of the oligos), the results for cTNNI was not showed.  For detail, please refer to PAGE 4, line 119 of the revised manuscript and revised Figure S1 of the supplementary materials.

Comment 3:

The Authors should provide in material and methods QC analysis of RNA and DNAhow Pathway analysis was realized which algorithm has been used for that; and clarify how contractile function analysis in terms of cell shortening was evaluated (describe in detail how videos were analyzed).

Response to comments 3:

As suggested, for the RNA sequencing, we have added the details for QC and pathway analysis to the material and method section. For the genomic DNA methylation, as it was conducted following the previous publications, we have provided the citations (for detail, please see Page 12, line 393).

For the contractile function analysis, we have added the detail descriptions on how the change of cell length was measured, for detail, please see Page 11, line 367.

Comment 4:

I recommend the Authors to describe western blotting analysis, what program did you use to quantize the bands? It does not appear in the manuscript.

Response to comment 4:

As suggested, we have added the details of Western blot analysis in the method section of the revised manuscript. For the quantification, we used the image J software developed by national institute of health. For details please refer to Page 13, line 432.

Comment 5:

Finally, in addition to the above points, consider re-writing the conclusion part along with prospects. In this, the Authors should also discuss the real-life utility of this work and the author's future aims and objectives with these as-prepared materials as it should also be mentioned because science should be done to get real outcomes, not just publication-based outcomes. I believe that this is intended to improve the quality of the manuscript that is deemed suitable for publication.

Response to the comment 5:

We agree with the reviewer that the conclusion should be in line with the real-life utility, as such, we have remove the parts related to neural dysfunction, and refer back to the cardiac dysfunctions that usually observe in Rett Syndrome patients. For details, please see Page 14, line 457.

Minor issues"

English needs to be improved, there are some words and expressions that need to be revised, for example add a single space after a period or any other punctuation mark you use to end a sentence in the line 361 (iPSC-CMscaused), line 401 ([28].To).

Response to the minor issues:

We are sorry about the typo errors. In fact, the manuscript have been submitted to English editing, and the space/typo error have been fixed in the revised manuscript.

Round 2

Reviewer 2 Report

First of all, I would like to appreciate the authors for the revision of the manuscript in accordance my comments and for the additional data that definitely improved the manuscript for publication. I believe that the manuscript it suitable for publication in its present form.